# Continuous sterane and phytane $\delta^{13}$C record reveals a substantial $p$CO$_2$ decline since the mid-Miocene

Caitlyn R. Witkowski [1,2] ✉, Anna S. von der Heydt [3], Paul J. Valdes [4], Marcel T. J. van der Meer [1], Stefan Schouten[1,5] & Jaap S. Sinninghe Damsté [1,5]

Constraining the relationship between temperature and atmospheric concentrations of carbon dioxide ($p$CO$_2$) is essential to model near-future climate. Here, we reconstruct $p$CO$_2$ values over the past 15 million years (Myr), providing a series of analogues for possible near-future temperatures and $p$CO$_2$, from a single continuous site (DSDP Site 467, California coast). We reconstruct $p$CO$_2$ values using sterane and phytane, compounds that many phytoplankton produce and then become fossilised in sediment. From 15.0-0.3 Myr ago, our reconstructed $p$CO$_2$ values steadily decline from 650 ± 150 to 280 ± 75 ppmv, mirroring global temperature decline. Using our new range of $p$CO$_2$ values, we calculate average Earth system sensitivity and equilibrium climate sensitivity, resulting in 13.9 °C and 7.2 °C per doubling of $p$CO$_2$, respectively. These values are significantly higher than IPCC global warming estimations, consistent or higher than some recent state-of-the-art climate models, and consistent with other proxy-based estimates.

Defining the relationship between the atmospheric concentration of carbon dioxide ($p$CO$_2$) and temperature is essential for understanding present environmental changes and modelling future climate trends. Geologic data can provide critical context, as well as possible analogues, for our future. For example, as compared to today's global annual temperatures of 14.5 °C[1], the middle Miocene (ca. 15 million years ago; Ma) was 18.4 °C[2–4], equivalent to that predicted for the year 2100 using the IPCC RCP8.5 scenario[5,6]. Thus, studying the past 15 million years (Myr) may provide a series of climate analogues relevant for possible near-future climates.

Over the past 15 million years, $p$CO$_2$ values have steadily declined, according to the latest $p$CO$_2$ compilation[7]. Within this revised compilation, however, estimates widely range both among proxies and within a single proxy (i.e., >500 ppmv difference) and include some unrealistically low values (i.e., <120 ppmv), providing the need for additional independent proxy records. Furthermore, our current understanding of the past 15 Myr is comprised of compilations[7–10] of much shorter intervals, e.g., the recent study by Brown et al. [11] covering 5–7 Ma; no single site covers the entirety of the past 15 Myr, which introduces concerns in stitching disparate sections together (e.g., regional and latitudinal influences). Thus, $p$CO$_2$ reconstructions over the past 15 Myr require further clarification from an additional independent proxy record from a single site that spans this whole period, especially given the importance of defining the relationship between $p$CO$_2$ and temperature for future climate scenarios.

Here, we aim to better define the relationship between $p$CO$_2$ and temperature from the mid Miocene to late Pleistocene by using a single site that covers this entire 15-Myr time interval, supplemented with additional sites that cover shorter timespans within the same time interval. We use a refreshed approach to estimate $p$CO$_2$ from the stable carbon isotopic fractionation that occurs during photosynthetic CO$_2$-fixation ($\varepsilon_p$). This isotopic fractionation occurs in photoautotrophs as their CO$_2$-fixing enzyme Rubisco selects $^{12}$C over $^{13}$C, resulting in isotopically more negative biomass than the dissolved inorganic carbon

[1]Department of Marine Microbiology and Biogeochemistry, NIOZ Royal Netherlands Institute for Sea Research, Den Burg (Texel) 1790AB, The Netherlands. [2]Schools of Earth Science and Chemistry, and the Cabot Institute, University of Bristol, Bristol BS8 1RJ, UK. [3]Institute for Marine and Atmospheric Research Utrecht, Department of Physics, Utrecht University, Utrecht 3584CC, The Netherlands. [4]School of Geographical Sciences and Cabot Institute, University of Bristol, Bristol BS8 1SS, UK. [5]Department of Geosciences, Utrecht University, Utrecht 3508 TA, The Netherlands. ✉e-mail: caitlyn.witkowski@bristol.ac.uk

source (e.g., growth water). The $\varepsilon_p$ framework assumes passive diffusion of $CO_{2[aq]}$ into photoautotroph cells and so the utilization of carbon-concentrating mechanisms by photoautotrophs in past oceans is an unavoidable limitation. However, three decades of field observations and laboratory cultures (e.g.[12–20],) have attributed ambient $pCO_2$ as the primary controlling factor for $\varepsilon_p$, in which higher $pCO_2$ results in higher $\varepsilon_p$ values. $\varepsilon_p$ can be calculated from the $\delta^{13}C$ of phytoplanktonic biomass corrected for the $\delta^{13}C$ of dissolved $CO_2$ ($CO_{2[aq]}$). $\varepsilon_p$ is then used to estimate the concentration of $CO_{2[aq]}$ via $CO_{2[aq]} = b/(\varepsilon_f - \varepsilon_p)$, where $\varepsilon_f$ is the maximum potential fractionation due to $CO_2$-fixation and $b$ represents carbon demand per supply for phytoplankton. Finally, $CO_{2[aq]}$ is converted to atmospheric $pCO_2$ via Henry's law assuming atmosphere-ocean equilibrium. This $b$ parameter represents physiological factors that may impact $CO_2$ uptake, e.g., growth rate, cell radius, and cell membrane permeability e.g.[21,22],. Recent culture experiments suggest that light energy, independent of its effect on growth rate, may also be an important control, with higher irradiance resulting in higher $\varepsilon_p$ values[15,19,23,24], but has not yet resulted in revisions to the $pCO_2$ proxy calculation. This recent work provides new and exciting questions to explore, making the application of $\varepsilon_p$ for reconstructing $pCO_2$ in different settings and time periods timely, especially in context of the latest $pCO_2$ compilation[7]. By using a diversity of independent proxy methodologies, ideally with consistent deployment of methods between groups, with honest and robust modelling of uncertainty within each system, we can then challenge and scrutinize persistent proxy outliers that range outside the uncertainty bands of multiple other proxies.

Whereas previous work has relied on the $\delta^{13}C$ of alkenones i.e., compounds produced by species within the Haptophyte clade, we expand the $\varepsilon_p$ approach to the $\delta^{13}C$ of general phytoplankton biomarkers (GPBs) i.e., compounds produced by the majority of photoautotrophs in sea surface waters and have subsequently become fossilized in marine sediments[25,26]. Several recent studies explored the potential of GPBs across a modern environmental transect from high $pCO_2$ near a naturally-occurring marine $CO_2$ seep towards control values in two drastically different geographic locations (i.e., off the coasts of Vulcano Island, Italy[25] and Shikine Island, Japan[26]). The applied GPBs known as phytol (i.e., the side-chain of the vital photoautotrophic pigment chlorophyll-a[27]) and cholesterol (i.e., a sterol that all eukaryotes synthesize or produce from ingested sterols with minimal isotopic fractionation[28,29]) demonstrate that mixed phytoplankton communities with varying cell sizes and growth rates still exhibit a strong isotopic response to $CO_{2[aq]}$. Phytol has further been tested across glacial-interglacial cycles, which suggest phytol reconstructions were within error of the ice core-based $CO_2$ records and showed nearly identical values as the alkenone-based reconstructions[30]. Phytol and sterols have less well-constrained sources than alkenones, possibly leading to more uncertainty in absolute $pCO_2$ estimates. However, because GPBs are produced by a large number of species, they may have several benefits over alkenones: (i) GPBs have greater spatial and temporal distribution (spanning at least 10x deeper in the geologic record)[31] and (ii) GPBs have the potential to curb species-specific concerns and environmental effects by averaging the whole phytoplankton community, while also being much more specific than the $\delta^{13}C$ of bulk sedimentary OM that has been used to this end[32]. Because GBPs are far more ubiquitous than alkenones, they provide more extensive coverage (both spatially and temporally) to generate a continuous record of $pCO_2$, overcoming a major hurdle with previous proxy-based $pCO_2$ reconstructions. As such, GPBs have the potential to span the Phanerozoic, whereas alkenones are limited to the Cenozoic, which would extend $\varepsilon_p$-based $pCO_2$ proxies by nearly ten-fold. Thus, this general phytoplankton biomarker approach is a promising tool to obtain paleo $pCO_2$ records.

Here, we estimate paleo $pCO_2$ from the $\delta^{13}C$ of GPBs over the past 15 Myr, as well as from the $\delta^{13}C$ of alkenones for proxy comparison.

## Results and discussion

### $\delta^{13}C$ values of GPBs over the past 15 Myr at DSDP Site 467

Marine sediments retrieved by drilling DSDP Site 467 (33.8495, −120.757833) off the coast of California are remarkably unique in that they contain OM-rich sediments over this entire timeframe (details in "Methods"). The most abundant and ubiquitous GPBs in these sediments are phytane, 5α-cholestane, 24-ethyl-5α-cholestane, and 24-methyl-5α-cholestane, the diagenetic products of our target GPBs (i.e., phytol and sterols). These GPBs occurred in organic sulphur (S)-rich macromolecules and were recovered by desulfurization. Because reduced inorganic S species rapidly react with functionalized labile lipids in anoxic surface sediments, S-bound molecules reflect in situ produced lipids. The low abundance of higher plant-derived long-chain $n$-alkanes and terpenoids from terrestrial inputs indicate that sedimentary OM at Site 467 is predominantly derived from marine sources. Thus, these phytane and steranes originate primarily from phytoplankton that are photosynthetically fixing dissolved $CO_2$ in the upper part of the water column[33]. Details on methods in Methods.

The $\delta^{13}C$ values of the GPBs steadily increase from 15.0 to 0.3 Ma (Fig. 1a; Supplementary Fig. 1; Supplementary Dataset 1–6). Here, we use a weighted average for the $\delta^{13}C$ of steranes, based on their fractional abundances (Supplementary Dataset 3–6). The $\delta^{13}C$ of phytane (ranging from −26.8 to −23.7‰) and weighted steranes (from −28.2 to −24.3‰) show statistically similar $\delta^{13}C$ trends throughout the record (Fig. 1, S3, S4), consistent with a similar general source, i.e., phytoplankton. These $\delta^{13}C$ records are consistent with the much shorter $\delta^{13}C$ records for GPBs (Supplementary Data 2) reported for the Monterey Formation at Naples Beach in the Santa Barbara basin[33] and Shell Beach in the Pismo basin[34], as well as for the $\delta^{13}C$ of phytane record from

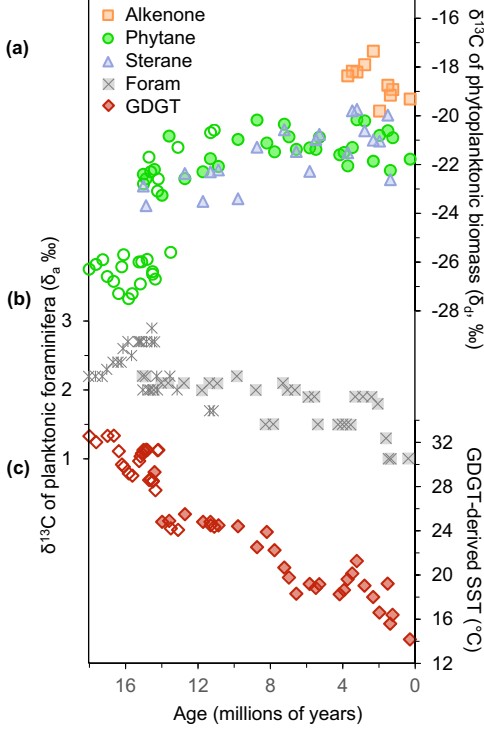

**Fig. 1 | Data covering the past 18 Myr used to calculate the atmospheric concentration of carbon dioxide ($pCO_2$) from phytoplankton biomarkers. a** $\delta^{13}C$ records of phytane (green circles), $C_{37:2}$ and $C_{37:3}$ alkenones (peach squares), and the weighted average of steranes (lavender triangle), (**b**) global compilation of $\delta^{13}C$ of planktonic foraminiferal shells[37], and (**c**) glycerol dibiphytanyl glycerol tetra-ethers (GDGTs)-derived sea surface temperatures (SSTs). For (**a**), (**b**), and (**c**), the closed symbols refer to the values for Site 467 and open symbols refer to additional sites. Source data are provided with this paper (Supplementary Data 1–9).

DSDP Site 608 in King's Trough in the eastern North Atlantic[3,35]. Although these three latter sections only span ca. 11–18 Ma, their corresponding results strongly suggest that the $\delta^{13}C$ records at DSDP Site 467 reflect a global (and not local) signal for $pCO_2$. Alkenones were present in only the most recent 4 Myr and their $\delta^{13}C$ values range from -21.4 to -23.8‰ (Fig. 1a; Supplementary Data 7), consistent with the $\delta^{13}C$ trends for the GBPs during that time.

## Calculations for $\varepsilon_p$ based on $\delta^{13}C$ values of GPBs

$\varepsilon_p$ is calculated from the $\delta^{13}C$ of phytoplanktonic biomass ($\delta_p$) and the $\delta^{13}C$ of aqueous carbon dioxide ($CO_{2[aq]}$) in the photic zone ($\delta_d$):

$$\varepsilon_p = 1000 \cdot [(\delta_d + 1000)/(\delta_p + 1000) - 1] \quad (1)$$

To determine $\delta_p$, we use the $\delta^{13}C$ of a phytoplanktonic biomarker lipid (Supplementary Fig. 1) corrected for the isotopic offset between the specific biomarker lipid and biomass (Fig. 1a). This offset is calculated based on compiled laboratory cultures, with the isotopic offset between biomass from phytane as 3.5 ± 1.3 SD ‰ based on its precursor phytol (compiled in ref. 36), steranes as 4.5 ± 3.0 SD ‰ based on their precursor sterols[37], and alkenones as 3.9 ± 0.4 SD ‰ (compiled in ref. 36). $\delta_d$ was estimated from a global compilation of $\delta^{13}C$ of planktonic foraminiferal shells[38] (Fig. 1b) and was corrected for the temperature-dependent carbon isotopic fractionation of $CO_{2[aq]}$ with respect to $HCO_3^-$ ref. 39. Sea surface temperatures (SST) were calculated using the $TEX_{86}$ proxy[40,41] based on the ratio of cyclopentane rings in glycerol dibiphytanyl glycerol tetraethers (GDGTs) in the same sediments as our GPBs (Fig. 1c; Supplementary Data S8) and assigned an uncertainty of ± 4 °C SD[40] caused by potential calibration errors. The estimated values for $\varepsilon_p$ are compiled in Supplementary Fig. 2.

$pCO_2$ was then calculated using[22,42]:

$$pCO_2 = [b/(\varepsilon_f - \varepsilon_p)]/K_0 \quad (2)$$

where $K_0$ reflects the Henry's Law constant that is used to convert $CO_{2[aq]}$ to $pCO_2$ based on temperature and salinity[43]. Within the brackets of Eq. (2), $\varepsilon_f$ reflects the maximum potential isotopic fractionation due to $CO_2$-fixation by the enzyme Rubisco[44], which is 26.5 ± 1.5 ‰ uniformly distributed uncertainty to reflect the full potential range reported in algal cultures (compiled in ref. 36). The $b$ parameter reflects species carbon demand per supply[22,42], which was back-calculated from bulk OM and phytol in a compilation of modern surface sediments worldwide[36] (i.e., 28 sites at different latitudes with known environmental parameters): the average is 168 ± 43 SD ‰ kg μM$^{-1}$. This value for $b$ is further supported by two phytol studies across two naturally occurring steep $CO_2$ gradients[25,26] and a phytol study in the equatorial Pacific Ocean[45], as well as previous paleoclimate studies using phytane where a $b$ value of 170‰ kg μM$^{-1}$ was used[46–48]. Thus, we apply 168 ‰ kg μM$^{-1}$ in all our calculations.

## $pCO_2$ estimations over the past 15 Ma

Expectedly, $\varepsilon_p$ calculated from the $\delta^{13}C$ of GPBs also all share similar values, ranges, and declining trends: 15.8 to 11.2‰ for phytane and 17.0 to 11.0‰ for steranes (Supplementary Fig. 2; Supplementary Dataset 1–6). This similarity among the GPBs is likewise reflected in the resulting $pCO_2$ estimations (Fig. 2). $pCO_2$ is highest at 15.0 Ma with values of 620 ppmv and 655 ppmv using phytane and steranes, respectively. These estimates tightly follow each other throughout the record, with the exception of the data point at 9.8 Ma, where phytane suggests a continued decline (435 ppmv) but steranes suggests a singular spike in $pCO_2$ (540 ppmv). By 8.7 Ma, phytane and sterane estimates converge (400 ppmv) for the rest of the record. Alkenone-based $pCO_2$ estimations, which could only be determined for the most recent 4 Myr period, where alkenones were present, were almost identical to those obtained with the GPBs.

The similarity in $pCO_2$ estimations between these GPBs is reassuring. However, we consider that these estimations may be influenced by the same factors, such as constraints on calculation parameters or upwelling. One of the more-difficult-to-constrain parameters in our calculation is factor $b$. Although it may change over time[21], it is not possible to constrain this value in most geologic settings; thus, maintaining $b$ as a constant is the most reasonable approach. Sensitivity tests demonstrate that the uncertainty within the $b$ value could lead up to a maximum of 25% change in $pCO_2$ estimation[36], which is still too small to account for the consistent decline over the studied time interval. Furthermore, the overlap between our GPB-based $pCO_2$ estimates with the more conventional alkenone-based $pCO_2$ estimates, for which substantial research on the $b$-value has been conducted[20,42], suggests that $b$ values for our GPB-based reconstructions are realistic.

Another potential factor to consider is change in upwelling intensity. Upwelling may mask the phytoplankton response to a changing $pCO_2$, bringing the ocean out of equilibrium with the atmosphere as upwelling brings more $^{13}C$-depleted $CO_{2[aq]}$ from cold bottom waters to the surface[45]. Radiolarian evidence[49] at the nearby ODP Site 1021 suggests enhanced coastal upwelling is unlikely in this region due to the timing of biosiliceous sedimentation in relation to known changes in the California Current. A potential increased production of North Atlantic Deep Water (NADW), the source of upwelling in this region, may have occurred between 11.5 to 10.0 Ma and from 7.6 to 6.5 Ma[49] (marked in Fig. 2). Notably, during these potential periods of increased NADW (Fig. 2), our biomarker-based $pCO_2$ values do not deviate from the overall downward trend. To further explore the potential impact of upwelling from a biomarker-perspective, we also analyzed the $C_{25}$ highly branched isoprenoids (HBI), a biomarker produced by specific diatoms that thrive in upwelling regions, as seen in the Arabian Sea over the past 0.3 Myr[50]. Although the diatom-produced $C_{25}$ HBI is present in our sediments, the $\delta^{13}C$ of the $C_{25}$ HBI varies greatly throughout the record and has no correlation with the $\delta^{13}C$ of the GPBs (Supplementary Fig. 1, 3). This lack of correlation suggests that these upwelling-related diatom species do not significantly contribute to the overall phytoplankton lipid pool and thus the effect of upwelling is likely minimal. We also considered the relative contribution of the $C_{28}$ sterane (24-methyl-5a-cholestane), which tends to be more dominant in diatoms; however, again, there is no relationship between the fractionation abundance of these diatom produced biomarkers with these periods of potential upwelling. Overall, although productivity changes or upwelling could play some role, these factors alone cannot account for our reconstructed ca. 350 ppmv decline in $pCO_2$ over 15 Myr.

Here, we put our results into context of earlier reports. We find that as compared with [19,51,52] the recently revised alkenone- and boron-based $pCO_2$ proxies compiled in Rae et al.[7] (Fig. 3a), our $pCO_2$ estimations follow similar trends and fall within error of estimate in Rae et al.[7], with absolute values closely matching throughout the record, especially the boron-based $pCO_2$ estimations. In the most recent 4 Myr of our record where our sediments contained alkenones, our $pCO_2$ values closely match the boron-based $pCO_2$ in Rae et al.[7], but tend to be slightly higher (ca. 50 ppmv) than the[7] alkenone-based $pCO_2$ estimations in Rae et al.[7]. This overall alignment with the most up-to-date records is promising; this suggests that our GPB-based values are likely producing reasonable $pCO_2$ estimations, but in this case, from a single continuous proxy record.

Furthermore, it is notable that our GPB-based $pCO_2$ estimates are consistent with the $pCO_2$ required by the majority of climate models in order to agree with the proxy-derived temperature estimates (and the generally accepted sensitivity to $pCO_2$). For the mid-Miocene Climate Optimum (17-15 Ma), two different versions of the National Centre for Atmospheric Research model (NCAR) indicate that $pCO_2$ needs to be within the range 460–580 ppmv[2,51], the Max-Planck Institute Earth System Model (MPI-ESM) suggests that $pCO_2$ should be around

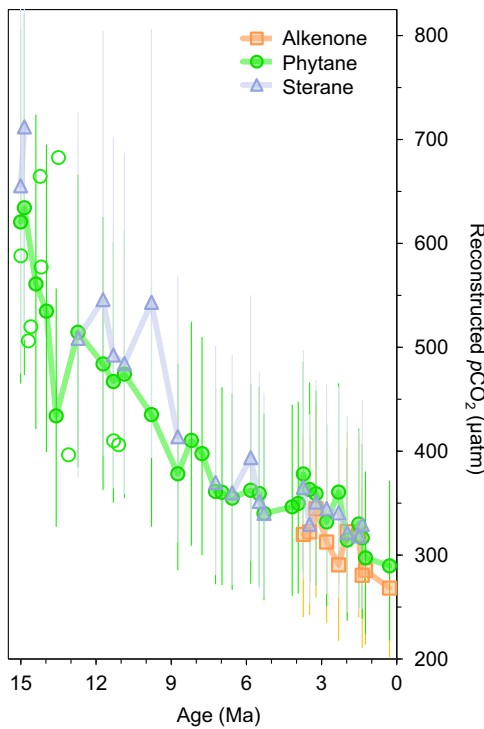

**Fig. 2 | Estimates for atmospheric concentration of carbon dioxide ($p$CO₂) derived from DSDP Site 467.** Covering the past 15 million years, $p$CO₂ calculations are based on the δ¹³C records of phytane (green circles), C₃₇:₂ and C₃₇:₃ alkenones (peach squares), and the weighted average of steranes (lavender triangle). Closed symbols represent Site 467, open symbols refer to additional sites. Error bars represent the one standard deviation based on Monte Carlo simulations that compound uncertainties for all input parameters. Shaded areas show two points of potentially-enhanced North Atlantic Deep Water. Source data are provided with this paper (Supplementary Data 1–9).

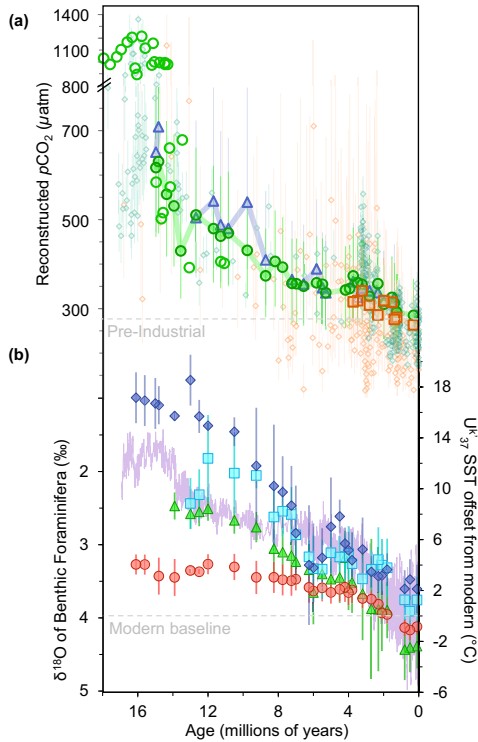

**Fig. 3 | Reconstructed the atmospheric concentration of carbon dioxide ($p$CO₂) and temperature estimates over the past 18 Ma. a** $p$CO₂ estimations based on the δ¹³C records of phytane (green circle, with closed symbols for Site DSDP 467, open symbols for additional sites), alkenones (peach squares), and the weighted average of steranes (lavender triangle). Error bars represent the one standard deviation based on Monte Carlo simulations that compound uncertainties for all input parameters. Previously compiled $p$CO₂ estimations[7] in diamond, alkenones (peach) and boron (teal). **b** Changes in $U^{K'}_{37}$-based mean annual sea surface temperature (SST) relative to modern[10] high-latitudes for the Northern Hemisphere (NH, blue), mid-latitudes (30-50°) for the NH (cyan) and Southern Hemisphere (SH, green), and tropics (30 °N-30 °S, red). Compiled δ¹⁸O derived from benthic foraminifera[54] (lavender) indicate bottom water temperature and build-up of continental ice volume. Source data are provided with this paper (Supplementary Data 1–9) or references therein.

720 ppmv[52], and Community Earth System Model (CESM1.0) requires around 800 ppmv[51]. Knorr et al.[53] did succeed in simulating the warmth of the mid-Miocene Climatic Optimum with relatively low $p$CO₂ but only by adopting large changes to the vegetation distribution, reducing planetary albedo, and having a strong positive water vapour feedback in their climate model. Therefore, our $p$CO₂ estimates are aligned with model-based interpretations of what reasonable $p$CO₂ should be across this period.

We also considered how our GPB-based $p$CO₂ estimations relate to proxy-based temperature. Based on visual comparison over this period (Fig. 3), it is clear that our $p$CO₂ record shows a similar declining trend as two different temperature proxies (Fig. 3b): the alkenone derived $U^{K'}_{37}$ proxy[10], which represents SST, and the δ¹⁸O derived from benthic foraminifera[54], which represents oceanic bottom water temperature, as well as the build-up of the continental ice volume. The remarkably similar trends of our $p$CO₂ estimates and independent temperature records from the Miocene through to the Pliocene therefore suggest that $p$CO₂ and temperature are closely coupled. A potential caveat for this observed correlation is the fact that SST is used twice in our paleo $p$CO₂ estimation (although notably, we have used the SST proxy TEX₈₆ to calculate our $p$CO₂, whereas our temperature comparisons in Fig. 3b are derived from different temperature proxies). Sensitivity tests over the Phanerozoic show negligible effects of SST on the first use in the equations, when calculating ε$_p$ (±0.5‰). In the second use of SST in the calculations (converting CO₂[aq] to $p$CO₂), sensitivity tests show that SST may potentially affect $p$CO₂ estimations up to ±50 ppmv[36]. Although a sizeable error, this potential ±50 ppmv is too small to suggest SST alone is driving the

declining trend ranging from ca. 630 to 280 ppmv over the entire record.

## Climate sensitivity

To explore the precise relationship between $p$CO₂ and global temperature, we calculated climate sensitivity, which refers to the impact of radiative forcing (which is primarily impacted by $p$CO₂) on temperature. Here, we calculate both Earth system sensitivity (ESS) and equilibrium climate sensitivity (ECS), respectively representing slow and fast climate feedback responses, using our new $p$CO₂ dataset from DSDP 467 (Table 1). We calculate an average sensitivity over 15 Myr and a range of $p$CO₂ values. Any variations over time could come from subsets of these points but given that there are a limited number (i.e., 30) of different (unequally spaced) time values, subdivisions and new regressions with all uncertainties would most likely give non-significant fits. The temporal dependence of climate sensitivity can only be determined with higher temporal resolution records for specific time intervals.

First, we calculate ESS, i.e., the response to CO₂ including (slow) climate feedbacks. ESS was estimated for each latitudinal region using a linear regression between the change in mean annual SST relative to modern SST (ΔSST) and radiative forcing due to CO₂ (ΔR$_{CO₂}$ [Wm⁻²]).

**Table 1 | Sensitivity of temperature to carbon dioxide ($CO_2$)**

| Region | Equilibrium Climate Sensitivity (ECS) | | | Earth System Sensitivity (ESS) | | |
|---|---|---|---|---|---|---|
| | K / Wm$^{-2}$ | r$^2$ | °C / 2×CO$_2$ | K / Wm$^{-2}$ | r$^2$ | °C / 2×CO$_2$ |
| NH High Latitude | 3.1 | 0.9 | 11.6 | 5.1 | 0.8 | 18.8 |
| NH Mid Latitude | 2.3 | 0.9 | 8.6 | 4.3 | 0.8 | 16.0 |
| SH Mid Latitude | 2.2 | 0.8 | 8.3 | 4.4 | 0.6 | 16.1 |
| Tropics[a] | 1.3 | 0.8 | 5.0 | 3.0 | 0.5 | 11.1 |
| Global | 1.9 | 0.8 | 7.2 | 3.8 | 0.6 | 13.9 |

[a]$U^{K'}_{37}$ ratio at these sites is reaching saturation (i.e., 1.00) from 15 to 8 Ma and SSTs derived from these reconstructed SSTs were omitted in our calculation.
Estimated Equilibrium Climate Sensitivity (ECS, fast feedbacks) and Earth System Sensitivity (ESS, slow feedbacks) over 15.0–0.3 Ma. ECS and ESS are expressed in temperature per radiative forcing (K/Wm$^{-2}$), the units typically used[64] for $S_{[CO2,X,Y]}$, and temperature change per doubling in $CO_2$ (°C/2×CO$_2$) based on their linear fit with errors in x and y. Calculations are shown by latitudinal region, including northern hemisphere (NH) and southern hemisphere (SH). Global refers to the weight by percent-area for the Earth: tropics (30°N-30°S, 50.0%), mid-latitudes (30–60°, 36.6%), and high latitudes (60–90°, 13.4%).

$\Delta$SST is based on the $U^{K'}_{37}$ proxy for SST, which were compiled[10] by latitude and hemisphere into 0.125 Ma bins (Fig. 3b) and linearly interpolated for the age of our sediments (Fig. 4). Tropical SSTs may be underestimated from 15 to 8 Ma, given that the $U^{K'}_{37}$ ratio is approaching saturation at these sites, and so to maintain consistency in proxy method but avoid bias in the tropic SSTs, we have not included $U^{K'}_{37}$ values when it approaches 1.0 in our calculations. $\Delta R_{CO2}$ was calculated using only the phytane-based $p$CO$_2$ record of DSDP 467, given that the GPBs and alkenones show similar $p$CO$_2$ values throughout this record and because the phytane record is most complete. Furthermore, phytane has yielded secular trends in $p$CO$_2$ comparable to other proxies in the Cretaceous[31,48] and over the Phanerozoic[36]. Monte Carlo simulations were used to propagate uncertainty for each equation parameter in these calculations[36].

The resulting ESS shows 18.8 °C (per $CO_2$-doubling) for NH high latitudes, 16.0 °C for the mid-latitudes, and 11.1 °C for the tropics (Supplementary Fig. 5), with respective values in K/Wm$^{-2}$ shown in Table 1 and are based on the slope of a linear fit of the data. When we weigh each sensitivity by the percent-area for the Earth: tropics (30°N-30°S, 50.0%), mid-latitudes (30–60°, 36.6%), and high latitudes (60–90°, 13.4%), our global average ESS amounts to 13.9 °C per doubling of $CO_2$. These values are considerably higher than the global ESS of 2.2 to 5.6 °C per $CO_2$-doubling calculated for the Plio-Pleistocene[55], although ESS from the same data (taking into account individual shifts in time) has been estimated as 9.0 ± 2.7 °C per $CO_2$-doubling (68% confidence level)[56,57] which is more consistent with our values, even though we use a similar approach (long-time average) as ref. 55. Our mid- and high-latitude ESS estimates suggest significant polar amplification despite generally less-than-present ice cover. Recent modelling efforts have highlighted the importance of cloud feedbacks in explaining very high polar sensitivity[58,59]. Even in largely ice-free climates of the Cenozoic, models suggest strong polar amplification due to cloud and land-surface feedbacks[60,61]. That said, the Cenozoic CO2PIP[7] estimations for ESS exceed ca. 8 °C per $CO_2$-doubling for the past 20 Myr, reaching ca. 13 °C per $CO_2$-doubling in the early Cenozoic.

Next, we calculate ECS, i.e., fast climate feedback and the quantity generally used in policy discussions. Given that our record spans 15.0 to 0.3 Ma, during which there is large variability in ice sheet coverage, we additionally consider radiative forcing due to land ice change ($\Delta R_{LI}$) based on earlier work[55,62,63], with results shown in Fig. 4. ECS was determined by a linear regression of $\Delta$SST versus $\Delta R_{CO2+LI}$ and is estimated to be 11.6 °C (per $CO_2$ doubling) for NH high latitudes, 8.6 °C for the mid-latitudes, and 5.0 °C for the

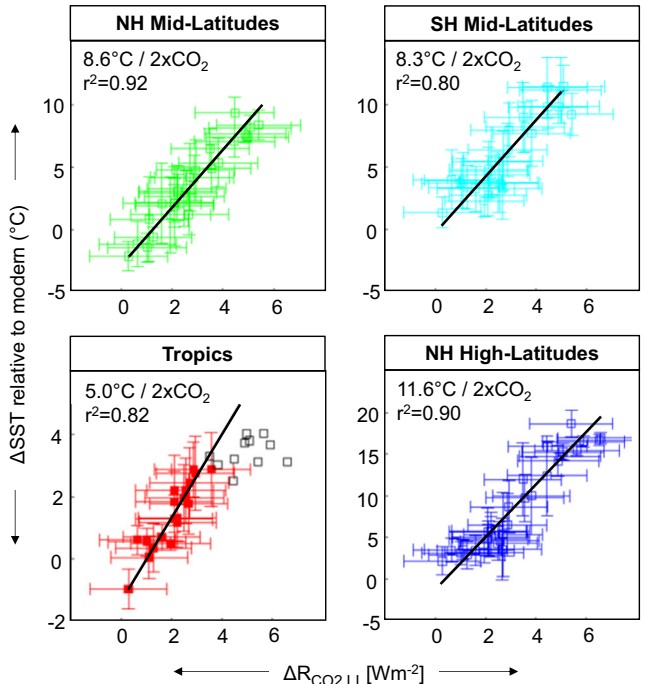

**Fig. 4 | Relationship between radiative forcing due to carbon dioxide ($CO_2$) and temperature from 15.0–0.3 Ma.** Y-axis: $U^{K'}_{37}$-based sea surface temperature (SST) changes relative to the modern mean annual SST at each site[10]: northern hemisphere (NH) mid-latitudes (green), southern hemisphere (SH) mid-latitudes (cyan), tropics (red), and NH high latitudes (blue). Error bars represent the one standard deviation based on Monte Carlo simulations that compound uncertainties for all input parameters. *Note that the $U^{K'}_{37}$ ratio approaching saturation beyond 8 Ma in the tropics cannot be used for a good computation of climate sensitivity, thus data > 8 million years (open black squares) are plotted but removed from fit. X-axis: Radiative forcing due to $CO_2$ and land-ice ($\Delta R_{CO2,LI}$), where $CO_2$ estimations are derived from the general phytoplankton biomarker phytane. Top left corners show equilibrium climate sensitivity change in temperature per doubling of $CO_2$ based on the slope of a linear fit of the proxy-based data. Source data are provided with this paper (Supplementary Data 2) and references therein.

tropics, with respective values in K/Wm$^{-2}$ and r$^2$ values shown in Table 1. When we again weigh each sensitivity by the percent-area for the Earth, our global average ECS is 7.2 °C per doubling of $CO_2$, much higher than the most recent IPCC estimates of 2.3 to 4.5 °C[6] and consistent with some of the latest state-of-the-art models which suggest ca. 5.2 °C[64]. It should be noted that our ECS is not the same as the ECS used by the IPCC, given that it represents specific climate sensitivity $S_{[CO2,LI]}$ (i.e., ESS corrected for potential slow land ice feedback) and does not consider changes in other greenhouse gases (e.g., methane), paleogeography, nor solar luminosity; we are currently unable to conduct these additional considerations[65]. The impact of additional methane and water would bring down ECS, which likely explains why paleo ECS is generally higher than modern models.

Our work represents the application of general phytoplankton biomarkers (GPB) to reconstruct $p$CO$_2$ values, offering a refreshed approach to the $p$CO$_2$ proxy based on photosynthetic isotopic fraction that may enable reconstructions over longer timescales where other existing proxies are lacking (e.g., the Phanerozoic). Our reconstructed $p$CO$_2$ values across the past 15 million years suggest Earth system sensitivity averages 13.9 °C per doubling of $p$CO$_2$ and equilibrium climate sensitivity averages 7.2 °C per doubling of $p$CO$_2$. Although these values are significantly higher than IPCC global warming estimations, they are consistent or higher than some recent state-of-the-art climate models and consistent with other proxy-based estimates.

## Methods

### Study site

Site 467 (33.8495, -120.757833) was collected by Deep Sea Drilling Project Leg 63 at the San Miguel Gap off the coast of California, USA. The total length of the core section is 1041.5 m and 426.3 m were recovered. The core has the best-preserved organic matter of Leg 63[66], likely due to incorporation of abiotic sulphur species into labile functionalized lipids, a process which occurs rapidly during very early diagenesis[67–69]. The age model is based on diatom, coccolith, and radiolarian events[66], which we have revised for every reported species using first and last occurrence, checked against the most up-to-date period tie points[54], and reported alongside mbsf from the core (details and references within Supplementary Data 9). The present-day oceanic regime of this region comprises of the California Current, a part of the North Pacific subtropical gyre which carries cold, fresher surface water from the North Pacific into the warmer, more saline surface water of the subtropical regions. Over long timescales, orbital forces impact the latitudinal changes, strength, and mean transport of the California Current flow.

### Analytical methodology

Thirty-five marine sediments, depths ranging from 9 to 1038 mbsf, were sampled approximately every 30 m from the Site 467 core (Dataset S1). 15–20 g of homogenized sediments were extracted on a Dionex 250 accelerated solvent extractor at 100 °C, $7.6 \times 10^6$ Pa, using dichloromethane (DCM):methanol (MeOH) (9:1 v/v) and the extracts were dried over $Na_2SO_4$. The extracts were eluted over an alumina packed column and separated into an apolar (hexane:DCM, 9:1 v/v), ketone (DCM), and a polar fraction (DCM:MeOH, 1:1 v/v). Polar fractions were desulfurized using Raney-nickel, eluted over an alumina packed column into an apolar fraction (hexane:DCM, 9:1 v/v) and hydrogenated using acetic acid and platinum oxide[67,68]. These were left over night and then cleaned over a small column of magnesium sulphate and sodium carbonate with DCM. To obtain baseline separation of the targeted biomarkers, *n*-alkanes were removed using vacuum-oven prepared 5 Å molecular sieve added to the samples, dissolved in cyclohexane, and left overnight; the supernatant was then removed and analyzed.

An Agilent 7890 A gas chromatograph-mass spectrometer (GC-MS) was used to identify GPBs (i.e., phytane and steranes), and the $C_{25}$ HBI alkane in the resulting apolar fraction from the desulfurized polar fraction, as well as alkenones in the ketone fractions. An Agilent 7890B GC with flame ion detector (FID) was used to determine compound quantities prior to injection on a Thermo Trace 1310 GC coupled to a Thermo Delta V-isotope ratio mass spectrometer (IRMS). GC-MS, GC-FID, and GC-IRMS measurements were conducted on a CP-Sil 5 column (25 m × 0.32 mm; $d_f$ 0.12 μm). GC-MS and GC-FID used constant pressure and IRMS used constant flow of He carrier gas. All three instruments used the same GC program with starting oven temperatures of 70 °C ramped at 20 °C/min to 130 °C and then ramped at 4 °C/min to 320 °C for 10 min. For IRMS measurements, a standard with *n*-alkanes ($C_{20}$ and $C_{24}$) with known isotopic values (−32.7 and −27.0‰, respectively) was run at the start of each day and then co-injected with samples to monitor the integrity of the instrument (within 0.5‰). At the start of each day, the IRMS underwent an oxidation sequence for 10 min, He backflushed after oxidation for 5 min, and conditioning line purged for 5 min; a shorter version of this sequence is conducted in a post-sample seed oxidation, which includes 2 min oxidation, 2 min He backflush, and 2 min purge conditioning line.

An Agilent 1260 ultra-high-performance liquid chromatography (UHPLC) coupled to a 6130 quadrupole MSD in selected ion monitoring mode was used to identify and integrate glycerol dibiphytanyl glycerol tetraethers (GDGTs) in the polar fraction. Separation was achieved on two UHPLC silica columns (BEH HILIC columns, 2.1 × 150 mm, 1.7 μm; Waters) in series, fitted with a 2.1 × 5 mm precolumn of the same material (Waters) and maintained at 30 °C

according to previously established methods[69–71]. With these GDGTs, we then apply the SST proxy known as $TEX_{86}$ (TetraEther indeX of tetraethers consisting of 86 carbon atoms), where the number of cyclopentane moieties increases along with SST[40]. Here, we use the modified version known as $TEX_{86}$-H, modified for (sub)tropical oceans and greenhouse periods where the function excludes crenarchaeol regio-isomer for (sub)polar oceans[40,72,73]. Because several minor isoGDGT were below the detection level in the deepest part of the studied section, it was not possible to obtain $TEX_{86}$ values. To accommodate for this, we compared the overall records from Site 467 to the $TEX_{86}$ values at Site 608 at the same latitude, as well as $U_{K}^{'37}$ values from the nearby Site 1010 (directly south of DSDP Site 467) and Site 1021 (directly north of DSDP 467); all four sites have near-identical SST values throughout the past 15 Myr, so we use these other sites to linearly extrapolate the several missing SSTs at Site 467. All raw data is available in Supplementary Data 8.

### Estimating climate sensitivity

ESS was then estimated for each latitude region using a linear regression of radiative forcing due to $CO_2$ ($\Delta R_{CO2}$) versus ΔSST. $\Delta R_{CO2}$ was calculated using:

$$\Delta R_{CO2} = [a_1(C - C0)^2 + b_1|C - C_0| + c_1 N_0 + 5.36] \times \ln(C/C_0) \quad (3)$$

where $C$ is $pCO_2$ at the time of forcing (our phytane-based $pCO_2$), $C_0$ is a reference $pCO_2$ (280 ppm), $N_0$ is the average concentration of $N_2O$, and the constant coefficients $a_1$, $b_1$, and $c_1$ are $2.4 \times 10^{-7}$, $7.2 \times 10^{-4}$, and $2.1 \times 10^{-4}$ $Wm^{-2}$ $ppm^{-1}$, respectively, based on previously established methods[72]. For ΔSST, we used the previously compiled[10] $U^{K'}_{37}$-based SSTs (expressed relative to the modern SST at each site) by latitude: northern hemisphere (NH) high-latitudes (Ocean Drilling Program Sites 883, 907, 982, 983), NH mid-latitudes (ODP Sites 1010, 1021, 1208), southern hemisphere (SH) mid-latitudes (ODP Sites 594, 1085, 1088, 1125), and tropics (ODP Sites 722, 846, 850, 1241, U1338) from the $U^{K'}_{37}$-proxy.

Given that this record spans 15.0 to 0.3 Ma, during which the ice sheet cover varied to a large extent, we also consider radiative forcing due to land ice change ($\Delta R_{LI}$) which we estimated by multiplying reconstructed sea level (m) by 0.0308 $Wm^{-3}$, based on earlier work[64,74]. Sea level over the last 16 Ma is estimated at a few instances (0 Ma = 0 m change in sea level relative to present[72] and 3.2 Ma = 24 m, 10.0 Ma = 67 m, 14.9 Ma = 66 m, and 19.5 Ma = 105 m)[73] and then linearly interpolated. ECS is then approximated by the specific climate sensitivity $S_{[CO2,LI]}$ (nomenclature is in Palaeosens[62]), which we determine by a linear regression of SST anomaly versus $\Delta R_{CO2} + \Delta R_{LI}$.

## Data availability

All data are generated and used in this study are available in the main text and/or the Supplementary Data 1–9.

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

## Acknowledgements

We thank Vittoria Lauretano for researching updates to the age model, Jort Ossebaar, Alle Tjipke Hoekstra, and Ronald van Bommel at the NIOZ for technical support, and Jack Middelburg, Heather Stoll, and Roderik van der Wal for scientific input. This study received funding from the Netherlands Earth System Science Centre (NESSC) through a gravitation grant (024.002.001) to JSSD and SS from the Dutch Ministry for Education, Culture and Science. AvdH and PJV acknowledge support from the TiPES project ('Tipping Points in the Earth System') from the European Union's Horizon 2020 research and innovation programme under

grant agreement no. 820970. CRW is supported by the Royal Society Dorothy Hodgkin Fellowship (DHF\R1\221014).

## Author contributions

C.R.W., S.S. and J.S.S.D. designed the study and wrote the manuscript. C.R.W. and MvdM analysed samples. ASvdH and P.J.V. conducted climate sensitivity analysis. All authors interpreted the data and contributed to the manuscript.

## Competing interests

The authors declare no competing interests.
