## [Peer Review File · Nature Communications]

Continuous sterane and phytane $\delta^{13}\text{C}$ record reveals a substantial pCO_2 decline since the mid-MioceneReviewer #1 (Remarks to the Author):

Witkowski and colleagues present a new record of CO₂ over the past 15 Myrs and pair these data with compiled temperature data in order to estimate climate sensitivity. It is a strong and well-written study that will be of interest to many. I start my review with some larger items, followed by a longer list of finer-level comments.

1) The authors describe at some length the strengths and weaknesses of their proxy. I encourage them to push further on this. My colleagues who specialize in the alkenone CO₂ proxy say that the proxy is undergoing a lot of growing pains, both in terms of understanding the mechanistic basis of the proxy but also in terms of parameterizing key variables related to growth rate (b, cell size, etc.). Many of the most recent alkenone papers publish a 'family' of plausible CO₂ curves. I'm not necessarily advocating such an approach here, but the bottom line is that whatever one thinks of the alkenone method, the phytane method is less robust (not as accurate, not as precise). It has less taxonomic control and incorporates fewer variables (the key advantage is that it is more widely applicable in time and space). There are good reasons why folks moved away from the phytane-style approach (like in Freeman and Hayes 1990) to alkenones. Now we've come full circle, in a way. The relative trends in reconstructed CO₂ should be quite robust, much more so than the absolute values. I encourage the authors to think about their data in these terms. For climate sensitivity, at least, relative changes are still extremely helpful!

2) Please describe how you combined the CO₂ and temperature data to create the individual data points in Figure 3. The CO₂ and temperature data in Figure 2 are—by my eye—diachronous, underscoring the importance of how you went about constructing the data points in Figure 3.

3) The 'global' climate sensitivities for ECS and ESS (5.0 and 11.1 oC per CO₂ doubling) seem to be on the high end. That is, not "consistent" (line 10) with many existing estimates. For example, your 5.0 oC ECS falls outside the 95% confidence window from the IPCC (1.5-4.5 oC). In addition, your estimates are based on surface temperatures of tropical oceans. The global ECS will be higher both because higher latitudes and land surfaces have higher ECS values. The newest IPCC report probably has the most up-to-date amplification factor between tropical oceans and the globe for the present-day Earth.

Your ESS value (11 oC) is consistent with some of the other proxy-constrained estimates of ESS. But if memory serves me right, ESS values derived from global climate models tend to be lower—on the order of 50% higher than ECS for this general time period (papers by Valdes, Lunt, and others).

4) I have concerns with the regression approach in Figure 3. Now, absolutely, the major advantage of this approach is that it smooths out the uncertainty of individual data points, which is admittedly quite large. But one key drawback is that the time element stitching together your data has been completely lost. There is no way to tell if climate sensitivity has changed over the captured time interval. Moreover, using a regression can mask the true relationship between CO₂ and temperature. It is possible, to give one example, for the slopes of all time-adjacent datapoints to be higher (or lower) than the regression slope. Or, half are steeper and half are shallower. These kinds of things would be really important to know. Yes, the uncertainty in each of the slopes connecting time-adjacent points is high, but in aggregate they should be very useful. A second advantage to analyzing time-adjacent data is that changes in other radiative forcings (for example, due to changes in paleogeography, the brightening of the sun, plant evolution, etc., etc.) should be small. This is not necessarily the case when you apply a single regression to an entire data set.

I also have some concerns about the type of regression used, which I am assuming is an ordinary least-squares regression. This type of regression assumes that the x-dimension has no uncertainty. That is, one is minimizing the residuals only in the y-dimension. I wonder if a major-axis regression is more appropriate (or one of its cousins), which incorporates variance in both dimensions. In most cases, major axis regressions are steeper. I've looked into the literature a bit on this topic, and I didn't see an easy answer. I'll end by adding that this is yet another reason for avoiding the regression approach—if you instead analyze time-adjacent data points, there is only one way to draw a straight line between two points (!).

Other comments:

Line 10: Need to say that these estimates are for tropical latitudes (not global).

Lines 28-30: "The most recent pCO₂ compilation (7) suggests that pCO₂ did indeed change over this 15-Myr period, but values are poorly constrained, i.e., ranging from 280 to 1460 μ atm at a single time-point. Thus, pCO₂ over the past 15 Myr is no less confusing..." Yes, this is true, but most of the variability is across proxy methods. In most cases, individual studies across the MCO show a compelling fall in CO₂, in keeping with the patterns of the current study. It's just that the absolute values vary across studies. Thus, in this sense, the current study doesn't address/rectify the problem posed in the quotation. I would argue that the MCO is one of the best pre-Pleistocene examples we have where almost all of the CO₂ data tell the same story (in terms of relative change).

Line 98: Citation to source #9 doesn't seem right. May be good to double-check the numbering throughout.

Lines 169-171: Are you talking about the MCO time slice?

Line 179: I'm not sure why you assume SST is a good proxy for global temperature.

Lines 186-190: You should estimate CO₂ assuming a fixed temperature, and then compare this to Figure 1. Then you will know, for your study, how important the confounding influence of temperature is. It's not clear if reference 23 looked at the same ranges of temperature and CO₂, etc.

Line 223: And Royer (2016, Figure 1 and Table 1) as well.

Royer, D. L., 2016, Climate sensitivity in the geologic past: Annual Review of Earth and Planetary Sciences, v. 44, p. 277-293, doi: 10.1146/annurev-earth-100815-024150.

Lines 223-224: The numbers you cite for reference #52 are in units of oC per CO₂ doubling, not oC per W/m². And, these are numbers for a non-glacial state. During glacial periods (which includes the last 34 Myrs) ESS is twice as high, according to #52.

Line 500: The variable "N" should really be called "N_0" so that it is clear you are talking about present-day concentrations of N₂O (at least, that is what I infer you are doing). And, why stop at N₂O? Why not include other greenhouse gases like methane?

Line 503: The citation to source #66 can't be right.

Reviewer #2 (Remarks to the Author):

This is a well-constructed study, building on the group's work on new pCO₂ proxies based on phytane and sterane biomarkers. I have no issues with the findings or the science.

I do, however, take issue with the set-up of the paper, which is that there is big uncertainty in late Miocene to Recent CO₂ estimations that they have now resolved - see the abstract. Given that their new estimates look (they don't reference or plot this study) very similar in trend to the best recent review and compilation of Cenozoic CO₂ (Rae et al. 2021) this is simply a confirmation (which is useful) rather than the resolution of a major problem. Although the Rae paper looks somewhat higher in absolute values, which again might be why they didn't plot or engage with that study - this is just not good enough.

If they'd referenced Rae et al. 2021 and factored this into their manuscript, some novelty may have come across, but the fact that they didn't means either:

1) they are willfully ignoring the main paper in this area to be published in the last couple of years (and thought no one would notice), or:

2) they are not aware of it - which, given the author list, I really can't believe.

The results are not of a novelty or importance sufficient to be published in Nature Communications. And I'm quite disappointed in the authors, if willfully ignoring key literature is their approach to "doing science".

This paper is suitable for publication with minor revisions in a disciplinary journal (e.g. P&P or CoP), but please, in the rewrite, cite and integrate all the relevant literature.

Response to the reviewers:

We would like to thank both the reviewers and the editor for their comments and suggestions, which have undoubtedly improved this manuscript. We respond to each individual comment below in **bold font** and the line numbers refer to the “clean” version of the manuscript (without track changes).

Reviewer #1:

Witkowski and colleagues present a new record of CO₂ over the past 15 Myrs and pair these data with compiled temperature data in order to estimate climate sensitivity. It is a strong and well-written study that will be of interest to many. I start my review with some larger items, followed by a longer list of finer-level comments.

We thank the reviewer for their positive comments.

1) The authors describe at some length the strengths and weaknesses of their proxy. I encourage them to push further on this.

We have further expanded the discussion around the strengths and weaknesses of the proxy in the Introduction Line 46-62.

My colleagues who specialize in the alkenone CO₂ proxy say that the proxy is undergoing a lot of growing pains, both in terms of understanding the mechanistic basis of the proxy but also in terms of parameterizing key variables related to growth rate (b, cell size, etc.). Many of the most recent alkenone papers publish a ‘family’ of plausible CO₂ curves. I’m not necessarily advocating such an approach here, but the bottom line is that whatever one thinks of the alkenone method, the phytane method is less robust (not as accurate, not as precise). It has less taxonomic control and incorporates fewer variables (the key advantage is that it is more widely applicable in time and space). There are good reasons why folks moved away from the phytane-style approach (like in Freeman and Hayes 1990) to alkenones. Now we’ve come full circle, in a way. The relative trends in reconstructed CO₂ should be quite robust, much more so than the absolute values. I encourage the authors to think about their data in these terms. For climate sensitivity, at least, relative changes are still extremely helpful!

We more explicitly acknowledge the weakness in phytane versus alkenones on Line 80-84, “Phytol and sterols have less well-constrained sources than alkenones, possibly leading to more uncertainty in absolute pCO₂ estimates. However, general phytoplankton biomarkers are far more ubiquitous than alkenones, providing more extensive coverage (both spatially and temporally) to generate a continuous record of pCO₂, overcoming a hurdle with previous proxy-based pCO₂ reconstructions.” Interestingly, our alkenone-based, phytane-based, and sterane-based pCO₂ estimates yield near-identical absolute values when found in the same sediments. The benefit of using a general biomarker is clear: phytane and sterane appear throughout the section, whereas alkenones are confined to the final 4 million years of the record.

The broad paleo-pCO₂ community would agree that all pCO₂ proxies have issues; because of this (and not despite this) we should continue reconstructing pCO₂ in context of multiple proxies, where similar values provide higher confidence for pCO₂ estimates and unlike values identify areas that require further investigation. We acknowledge

that relative changes are more robust, and this is why we used climate sensitivity analysis based on relative change, not absolute numbers.

2) Please describe how you combined the CO₂ and temperature data to create the individual data points in Figure 3. The CO₂ and temperature data in Figure 2 are—by my eye—diachronous, underscoring the importance of how you went about constructing the data points in Figure 3.

The methods for Fig. 3 are described in the section “Climate sensitivity”, with additional details in the Supplemental Material in the section “Estimating climate sensitivity”. On the y-axis of Fig. 3, we show mean annual SST relative to modern SST (Δ SST) from Herbert et al. (2016), the same data displayed in Fig. 2b. On the x-axis of Fig. 3, we show radiative forcing due to CO₂ (ΔR_{CO_2} [Wm^{-2}]) which is calculated from our phytane-based pCO₂ estimates in Fig. 2a, with additional considerations for land ice change (Δ RLI).

3) The ‘global’ climate sensitivities for ECS and ESS (5.0 and 11.1 oC per CO₂ doubling) seem to be on the high end. That is, not “consistent” (line 10) with many existing estimates. For example, your 5.0 oC ECS falls outside the 95% confidence window from the IPCC (1.5-4.5 oC). In addition, your estimates are based on surface temperatures of tropical oceans. The global ECS will be higher both because higher latitudes and land surfaces have higher ECS values. The newest IPCC report probably has the most up-to-date amplification factor between tropical oceans and the globe for the present-day Earth. Your ESS value (11 oC) is consistent with some of the other proxy-constrained estimates of ESS. But if memory serves me right, ESS values derived from global climate models tend to be lower—on the order of 50% higher than ECS for this general time period (papers by Valdes, Lunt, and others).

We have revised the ECS and ESS paragraphs for clarity. We agree and now explicitly state that, “Our estimate of the tropical ECS of 5.0°C suggests that the global ECS is just above the most recent IPCC estimates of 2.3 to 4.5°C per CO₂-doubling (Sherwood et al., 2020) and consistent with the latest state-of-the-art models which suggest ca. 5.2°C (Forster et al., 2020).” Regarding ESS, there is no general range for ESS due to time-specific changes (e.g., land ice distribution, paleogeography). We have now expanded this section to include some examples from the Pliocene and Phanerozoic to illustrate this point.

4) I have concerns with the regression approach in Figure 3. Now, absolutely, the major advantage of this approach is that it smooths out the uncertainty of individual data points, which is admittedly quite large. But one key drawback is that the time element stitching together your data has been completely lost. There is no way to tell if climate sensitivity has changed over the captured time interval. Moreover, using a regression can mask the true relationship between CO₂ and temperature. It is possible, to give one example, for the slopes of all time-adjacent datapoints to be higher (or lower) than the regression slope. Or, half are steeper and half are shallower. These kinds of things would be really important to know. Yes, the uncertainty in each of the slopes connecting time-adjacent points is high, but in aggregate they should be very useful. A second advantage to analyzing time-adjacent data is that changes in other radiative forcings (for example, due to changes in paleogeography, the brightening of the sun, plant evolution, etc., etc.) should be small. This is not necessarily the case when you apply a single regression to an entire data set.

I also have some concerns about the type of regression used, which I am assuming is an ordinary least-squares regression. This type of regression assumes that the x-dimension has

no uncertainty. That is, one is minimizing the residuals only in the y-dimension. I wonder if a major-axis regression is more appropriate (or one of its cousins), which incorporates variance in both dimensions. In most cases, major axis regressions are steeper. I've looked into the literature a bit on this topic, and I didn't see an easy answer. I'll end by adding that this is yet another reason for avoiding the regression approach—if you instead analyze time-adjacent data points, there is only one way to draw a straight line between two points (!).

We had conducted a linear fit with errors in x and y (not ordinary least-squares regression), as described in the Supplement, which we now explicitly state in the main text line 225, Table 1 captions, and Fig. 3 captions.

Other comments:

Line 10: Need to say that these estimates are for tropical latitudes (not global).

We have added that these are for tropical latitudes.

Lines 28-30: “The most recent pCO₂ compilation (7) suggests that pCO₂ did indeed change over this 15-Myr period, but values are poorly constrained, i.e., ranging from 280 to 1460 μ atm at a single time-point. Thus, pCO₂ over the past 15 Myr is no less confusing...” Yes, this is true, but most of the variability is across proxy methods. In most cases, individual studies across the MCO show a compelling fall in CO₂, in keeping with the patterns of the current study. It's just that the absolute values vary across studies. Thus, in this sense, the current study doesn't address/rectify the problem posed in the quotation. I would argue that the MCO is one of the best pre-Pleistocene examples we have where almost all of the CO₂ data tell the same story (in terms of relative change).

We have revised this paragraph (Lines 28-39). First, we have added the most recent compilation (Rae et al., 2021) which shows a clearer decline in pCO₂. Notably, even this most up-to-date version shows wide ranges in values among proxies and within a single proxy (i.e., >500 μ atm difference) and some unrealistically low values (i.e., <120 μ atm). Here, we also add that, “no single site covers the entirety of the past 15 Myr, which introduces concerns in stitching disparate sections together (e.g., regional and latitudinal influences)”; this is a major advantage of our study here, in which we show a continuous reconstruction over the full 15-myrr timespan from a single site.

Line 98: Citation to source #9 doesn't seem right. May be good to double-check the numbering throughout.

We have checked/revise all references throughout the manuscript.

Lines 169-171: Are you talking about the MCO time slice?

Yes, we have now added “For the mid-Miocene Climate Optimum” at the start of the sentence (line 193).

Line 179: I'm not sure why you assume SST is a good proxy for global temperature.

We have removed the phrase “global temperatures” for clarity. We do not assume SST is a good proxy for global temperature, which is why we latitudinal ESS and ECS instead of global ESS and ECS.

Lines 186-190: You should estimate CO₂ assuming a fixed temperature, and then compare this to Figure 1. Then you will know, for your study, how important the confounding influence of temperature is. It's not clear if reference 23 looked at the same ranges of temperature and CO₂, etc.

To clarify, we have added “Phanerozoic” to this sentence. The influence of temperature on this proxy has been calculated in Witkowski et al. (2018, see Supplement) for the Phanerozoic, which includes a much larger range of CO₂ and temperature. Temperature impacts every marine-based CO₂ proxy, as it is a critical parameter to calculate Henry’s Law from dissolved CO₂ to atmospheric CO₂ concentrations.

Line 223: And Royer (2016, Figure 1 and Table 1) as well.

Royer, D. L., 2016, Climate sensitivity in the geologic past: Annual Review of Earth and Planetary Sciences, v. 44, p. 277-293, doi: 10.1146/annurev-earth-100815-024150.

We have added Royer (2016).

Lines 223-224: The numbers you cite for reference #52 are in units of oC per CO₂ doubling, not oC per W/m². And, these are numbers for a non-glacial state. During glacial periods (which includes the last 34 Myrs) ESS is twice as high, according to #52.

In our manuscript there has been some exchange of numbers (and units), which created the confusion. We have now sorted and corrected all numbers and mention ESS and ECS in the text consistently in units of °C/2xCO₂. According to the convention defined in Rohling et al. (2012), the numbers in units of K/Wm⁻² should be named S_[CO₂,x,y] which we have now introduced in our paper. Nevertheless, we give values in both units in Table 1. In fact, we find a higher ESS than Wang et al. (2021) and Martínez-Botí et al. (2015), but within the range of Royer (2016).

Line 500: The variable “N” should really be called “N_0” so that it is clear you are talking about present-day concentrations of N₂O (at least, that is what I infer you are doing). And, why stop at N₂O? Why not include other greenhouse gases like methane?

Indeed, the equation used to calculate the radiative forcing from CO₂ changes should better be named N_0 as it is the present-day value for N₂O. We have changed that in the text (Line 568-570). According to this equation, we could take into account other greenhouse gases such as N₂O, but as we have no palaeo-records for this, we restrict the calculation only to CO₂.

Line 503: The citation to source #66 can’t be right.

We have checked/revise all references throughout the manuscript.

Reviewer #2:

This is a well-constructed study, building on the group's work on new pCO₂ proxies based on phytane and sterane biomarkers. I have no issues with the findings or the science.

We thank the reviewer for the positive comments.

I do, however, take issue with the set-up of the paper, which is that there is big uncertainty in late Miocene to Recent CO₂ estimations that they have now resolved - see the abstract. Given that their new estimates look (they don't reference or plot this study) very similar in trend to the best recent review and compilation of Cenozoic CO₂ (Rae et al. 2021) this is simply a confirmation (which is useful) rather than the resolution of a major problem. Although the Rae paper looks somewhat higher in absolute values, which again might be why they didn't plot or engage with that study - this is just not good enough.

If they'd referenced Rae et al. 2021 and factored this into their manuscript, some novelty may have come across, but the fact that they didn't means either:

1) they are willfully ignoring the main paper in this area to be published in the last couple of years (and thought no one would notice), or:

2) they are not aware of it - which, given the author list, I really can't believe.

The results are not of a novelty or importance sufficient to be published in Nature Communications. And I'm quite disappointed in the authors, if willfully ignoring key literature is their approach to "doing science".

This paper is suitable for publication with minor revisions in a disciplinary journal (e.g. P&P or CoP), but please, in the rewrite, cite and integrate all the relevant literature.

We agree that Rae et al. (2021) is the best recent review and compilation of Cenozoic CO₂ for boron-based and alkenone-based proxies and are grateful for the opportunity to include this study in our current manuscript. We have updated this manuscript to include the Rae et al. (2021) paper in the Introduction (Line 28-39), Discussion (Line 182-189), and Fig. 2. We have moved the older literature to the Supplement.

With respect, neither of the reviewers' assumptions are true. Instead, our manuscript was actively in review when Rae et al., 2021 was published; we hadn't updated our manuscript for Rae et al. prior to resubmission to Nature Communications, where our manuscript has been under this current round of reviews since the beginning of 2022. We are grateful for the chance to correct this.

Reviewer #1 (Remarks to the Author):

The authors have addressed many of my concerns, but some remain. I start with the two larger issues, followed by more minor comments.

1) I do not agree—for two reasons—with the authors' interpretation for a constant climate sensitivity (in title, abstract, and main text). First, their interpreted ECS and ESS values are for the tropics, which will be lower than the global-temperature-based estimates they are comparing against because half of the world's surface area is not in the tropics, and one-third of the world's surface area is on land. Both of these categories have higher regional climate sensitivities, meaning global-based climate sensitivities calculated from the authors' data will be higher than their reported tropical estimate of 5.0 oC. IPCC AR6 reports an ECS of 2-5 oC at 90% confidence. After adjusting the authors' estimate of 5 oC to account for the tropical bias, I wouldn't say that this is only "slightly higher" (remembering that the IPCC considers 5 oC only 5% likely). For ESS, there is a large body of literature pointing to a ~6 oC ESS for the Pleistocene, which is the closest we can get to the notion of "present-day". Is 11 oC "consistent" with 6 oC ??? I find it interesting that your estimates of ECS and ESS are higher than the present-day—indeed, it makes for an interesting story!

Second, the authors' data analysis is not sufficient to test if climate sensitivity has remained constant "since the mid-Miocene" (to use the language from the Title). The authors estimate climate sensitivity by calculating the regression slope through a cross-plot of radiative forcing vs. temperature. This tells you the *average* climate sensitivity for the chosen interval (15 Ma to near-present-day). It is possible that the average climate sensitivity is close to the present-day value but that it fluctuated a lot within the chosen interval. These *temporal* associations are lost when relying on a single regression. What does Figure 3 look like when you include the temporal associations (in other words, add lines connecting the time-adjacent data points)? This will give you a lot more information about if climate sensitivity has indeed remained constant.

2) The authors have not adequately addressed my question about combining the CO2 radiative forcing and temperature data in Figure 2 to make the cross-plots in Figure 3. To re-state: the ages of the CO2 and temperature data are not identical (at least not to my eye, looking at Figure 2). So, if this is true, how did you modify radiative forcing or temperature so that you could plot them directly against each other? If you simply took the temperature data closest in age to each single estimate of radiative forcing, this should be stated.

Line 16: *concentration* of CO2 (ppmv) is a much more useful concept in this context than partial pressure (for example, during the Miocene the total pressure of the atmosphere at sea level may have been somewhat different). And, most CO2 proxies are sensitive to concentration, not partial pressure (and so in contrast to what is said in line 24).

Line 25: This is a strawman argument. Foster et al. use a low-pass filter over a 400 million year long record, so it is not an apples-to-apples comparison to your record. If you look at the *raw* data in the Foster compilation (a better apples-to-apples comparison), there is a definite peak of ~600 ppm during the MCO. This peak is one of the best-resolved features of proxy-reconstructed Cenozoic CO2, present in multiple studies and multiple proxies. So it's odd (and wrong) to imply that the existing CO2 record is weak here.

Lines 244-245: Sure, but you use the same regression approach of Martinez-Boti, so the best comparison is with that study.

Lines 257-259: It's important to say here, in the main text, that your ECS is not the same as the ECS's in the IPCC; most importantly, you don't consider the potential for changes in other GHGs. You also don't consider changes in paleogeography or solar luminosity (short-term wiggles or long-term secular increase).

Reviewer #2 (Remarks to the Author):

I appreciate the efforts of the authors to now include the Rae et al. 2021 CO₂ compilation, but even if this manuscript had been submitted to another journal beforehand, when the manuscript was submitted to Nature Communications the Rae et al. 2021 paper was published, well known and should have been included in the original submission.

In the new introduction, lines 22-39, they include Rae et al. but still start from an argument of the Foster et al. compilation and the 'problem' of low middle Miocene estimated CO₂ concentrations. In my view they still need to accept that the Foster compilation is superseded and that there's no longer a significant middle Miocene problem. If they did this, I think they could make a better argument for the novelty of their approach and its importance – I think it's more significant as a new method that can extend CO₂ records further back in time than the alkenone CO₂ proxy (and a study that tests the robustness of this method).

I appreciate the work that has gone into the modern calibration and ground-truthing of this new CO₂ proxy approach, and also the detailed organic geochemistry to carefully extract and analyse these GPBs from DPDP 467 sediments – including the desulfurization steps – that are a significant benchmark and reference for such analyses. With more of a refocus on the importance (a revised introduction – see comments above) and details of the new methods (see comments below), I think this could be considered for publication.

Given the novelty of these records, I would like to see more of the primary data in the main text. At present they go straight to a CO₂ record in their first figure – it looks good and makes sense considering the long-term Neogene climate evolution, but the fact that it arrives "fully formed" as the first figure in the paper prompts questions as to the underlying assumptions and data – is it really as good as it looks (and I have serious concerns about the SST data – outlined below)? I would like to see the primary δ¹³C of the biomarkers, the long-term planktic foraminiferal δ¹³C compilation they use, the sea surface temperature records and the calculated Ep all plotted in the main text. Some of these are plotted in the SI but not all. All these data combine to generate the final CO₂ record, and I would like to see the trends, magnitudes of change and variability in these records in order to understand the components that contribute to the final CO₂ record. I would also like further justification of:

- The use of the global compilation of planktic foraminiferal δ¹³C – how was the uncertainty on this propagated through into CO₂ estimates? How well is this expected to represent local conditions at the study site? How do California Margin seawater DIC δ¹³C compare to the "global ocean" today?

- SST records - these should be plotted in the main text. How sensitive is the final CO₂ estimate to the choice of GDGT SST proxy? E.g. does the use of BAYSPAR substantially alter CO₂ records? I would like to see a sensitivity test with CO₂ curves plotted for at least the BAYSPAR SST estimations as well as the Kim et al. (2010) calibration of TEX₈₆. Checking the data tables I'm also not happy with two things:

1) The SST used for the CO₂ estimations (S1) say that they are Uk37 based temperatures not TEX₈₆ based temperatures. What is going on here? The calculations in the tables don't match the methodologies outlined in the text?

2) And further, the actual TEX₈₆ data is not available for 10 samples at the base of the record because GDGT abundances are too low. Where is this mentioned in the methods? And what did they do without TEX (although as noted in point 1 – it actually looks like it's Uk37 that is being used). If one of the global records for SST from Uk37 is being used (I can't see it being from the actual site – as there are no alkenones recorded in the older part of the section), then I am very concerned about circularity here – driving the CO₂ proxy with a global SST signal and then comparing it back against that same SST curve and finding a strong correlation.... This may not be what they are doing, but I'd like a very good explanation of what is happening here, as the

methods in the text don't seem to match the data tables. As with the failure to present the Rae et al. 2021 compilation, the jump straight to a finished CO₂ curve in the main text with none of the underlying data clearly shown in the main text, and then when the working is checked, to find it doesn't match the stated methods in the paper, and further to discover substantial chunks of (GDGT SST) data missing, is very concerning.

- The 'b' term. Although this is based on modern calibration studies, which look to be convergent on a similar number, I'm surprised at the "sensitivity" test of a "1% variability" in b which gives only a "1% change in CO₂" (lines 152 -156). This 1% variability doesn't even span the difference between the two figures they quote for b in the text (168 and 170 per mil kg μM⁻¹). I think those estimating CO₂ from the alkenone-based proxies would be over-joyed to know the 'b' term for the coccolithophores to within 1%. In the case of the alkenones it's clear that estimates of the 'b' term in the modern ocean do not translate into an accurate representation of the b term on even glacial-interglacial timescales (e.g. see discussion in Rae et al. 2021; Badger et al. 2020; Zhang et al 2019). In the case of the coccolithophore alkenone-producers, this variation in the b term is within relatively few species and just through the late Pleistocene. So, I am sceptical that a 1% variation in the b term is an appropriate sensitivity test for general (unconstrained) phytoplankton biomarkers on significantly longer (~15 Ma) timescales. Not only could there be significant evolutionary adaptation to long-term declining CO₂ concentrations (as seen in the coccolithophores – see Bolton, Stoll and many other papers), in the single site there are likely to be significant changes in dominant phytoplankton community, growth rates and size distributions. I do appreciate the efforts to test community change through the cross-comparison of the d13C data with biomarker indications for upwelling, but I don't agree with the assumption that the b term is well-constrained (<1% variation!) in this application or that it is stable on long-timescales.

Paragraph from Line 191 onwards – sounds like confirmation bias. That you get the answer for CO₂ from a proxy that fits with what the models need to reach Miocene temperatures is not the way to reassure yourselves that you have a robust CO₂ proxy – or that the models are robust. Keeping clear blue water between these two is important to provide confidence that they are independent, and we are properly testing model behaviour with the proxy data. Getting the 'right answer' is no guarantee that one's methods are sound, or, in this case, that one has actually arrived at the correct answer.

Line 213 – 215; why don't they do the sensitivity test with at least two common approaches to GDGT-based SSTs? This would give at least some representation of what (minimum) SST uncertainty in their records might translate to in terms of CO₂ rather than relying on other studies. I also don't understand why they don't plot their own SST record from this site as well as the existing alkenone compilations? – the local SST records may not be the most helpful for the Fig 2 comparison of CO₂ against global SST records, but the fact that the data is not plotted anywhere is not helpful. See also the comments above about which SST they actually have from the site, and which data they use to calculate CO₂.

Finally, if I were Reviewer 1, I would not be satisfied that the revised manuscript has properly engaged with their comments and concerns on several fronts. One aspect that overlaps with my comments above is the discussion and presentation of the 'b' term. As reviewer 1 notes, we seem to be coming "full circle" and somehow finding that phytane CO₂ records are better constrained than the alkenone CO₂ records. I agree that they might be more widely applicable, but like Reviewer 1 I'm sceptical that d13C fractionation factors from biomarkers that are more poorly constrained in terms of their biological source and growth environment can do better than ones that are well-constrained.

Reply to reviewer comments

Line numbers refer to the “track changes” version of the manuscript revisions.

Reviewer #1

The authors have addressed many of my concerns, but some remain. I start with the two larger issues, followed by more minor comments.

We thank the reviewer for the additional comments, which we address below.

1) I do not agree—for two reasons—with the authors’ interpretation for a constant climate sensitivity (in title, abstract, and main text). First, their interpreted ECS and ESS values are for the tropics, which will be lower than the global-temperature-based estimates they are comparing against because half of the world’s surface area is not in the tropics, and one-third of the world’s surface area is on land. Both of these categories have higher regional climate sensitivities, meaning global-based climate sensitivities calculated from the authors’ data will be higher than their reported tropical estimate of 5.0 °C. IPCC AR6 reports an ECS of 2-5 °C at 90% confidence. After adjusting the authors’ estimate of 5 °C to account for the tropical bias, I wouldn’t say that this is only “slightly higher” (remembering that the IPCC considers 5 °C only 5% likely). For ESS, there is a large body of literature pointing to a ~6 °C ESS for the Pleistocene, which is the closest we can get to the notion of “present-day”. Is 11 °C “consistent” with 6 °C ???? I find it interesting that your estimates of ECS and ESS are higher than the present-day—indeed, it makes for an interesting story!

We agree with the reviewer. We removed “constant climate sensitivity” from the manuscript, including the title, abstract, and main text.

We changed our phrasing to “significantly higher than the IPCC...” (Line 14-18) and “much higher than the most recent IPCC...” (Line 289).

We recalculated our ESS and ECS to consider a more global value (Lines 14-18, 268-270, and 287-288), where we weigh each sensitivity by the percent-area for the Earth: tropics (30°N-30°S, 50.0%), mid-latitudes (30-60°, 36.6%), and high latitudes (60-90°, 13.4%).

For example, the abstract revisions read as: “We calculate an average Earth system sensitivity and average equilibrium climate sensitivity over 15 Myr using our new range of $p\text{CO}_2$ values (weighed by percent-area for the Earth), resulting in 13.9 and 7.2°C per doubling of $p\text{CO}_2$, respectively. These values are consistent with other proxy-based estimates, consistent or higher than some recent state-of-the-art climate models, and significantly higher than IPCC global warming estimations.”

Second, the authors’ data analysis is not sufficient to test if climate sensitivity has remained constant “since the mid-Miocene” (to use the language from the Title). The authors estimate climate sensitivity by calculating the regression slope through a cross-plot of radiative forcing vs. temperature. This tells you the “average” climate sensitivity for the chosen interval (15 Ma to near-present-day). It is possible that the average climate sensitivity is close to the present-day value but that it fluctuated a lot within the chosen interval. These “temporal” associations are lost when relying on a single regression. What does Figure 3 look like when you include the temporal associations (in other words, add lines connecting the time-adjacent data points)? This will give you a lot more information about if climate sensitivity has indeed remained constant.

We agree with the reviewer that this shows the average climate sensitivity. In the abstract, we clarify that “We calculate an average Earth system sensitivity and average equilibrium climate sensitivity over 15 Myr...”

We have added Lines 245-250 to address the temporal associations, “We calculate an average sensitivity over 15 Myr and a range of $p\text{CO}_2$ values. Any variations over time could come from subsets of these points but given that there are a limited number (i.e., 30) of different (unequally spaced) time values, subdivisions and new regressions with all uncertainties would most likely give non-significant

fits. The temporal dependence of climate sensitivity can only be determined with higher temporal resolution records for specific time intervals.”

2) *The authors have not adequately addressed my question about combining the CO₂ radiative forcing and temperature data in Figure 2 to make the cross-plots in Figure 3. To re-state: the ages of the CO₂ and temperature data are not identical (at least not to my eye, looking at Figure 2). So, if this is true, how did you modify radiative forcing or temperature so that you could plot them directly against each other? If you simply took the temperature data closest in age to each single estimate of radiative forcing, this should be stated.*

We apologize for not adequately addressing your question. We have revised Line 254-255, “ Δ SST is based on the U^K₃₇ proxy for SST, which were compiled⁹ by latitude and hemisphere into 0.125 Ma bins (Fig. 3b) and linearly interpolated for the age of our sediments (Fig. 4).”

*Line 16: *concentration* of CO₂ (ppmv) is a much more useful concept in this context than partial pressure (for example, during the Miocene the total pressure of the atmosphere at sea level may have been somewhat different). And, most CO₂ proxies are sensitive to concentration, not partial pressure (and so in contrast to what is said in line 24).*

We have replaced all μ atm with ppmv and revised Line 22 “atmospheric concentration of carbon dioxide...”

*Line 25: This is a strawman argument. Foster et al. use a low-pass filter over a 400 million year long record, so it is not an apples-to-apples comparison to your record. If you look at the *raw* data in the Foster compilation (a better apples-to-apples comparison), there is a definite peak of ~600 ppm during the MCO. This peak is one of the best-resolved features of proxy-reconstructed Cenozoic CO₂, present in multiple studies and multiple proxies. So it's odd (and wrong) to imply that the existing CO₂ record is weak here.*

We agree with the reviewer. We were not referring to the MCO, but to the entirety of the past 15 million years (i.e., 15 Ma through to 0 Ma). The raw data in Foster et al. (2017) does indeed show data sparse intervals (e.g., 12-8 Ma). Regardless, we have removed Foster et al. (2017) from the Introduction given that it has been superseded by Rae et al. (2021) (see comments Reviewer #2).

Lines 244-245: Sure, but you use the same regression approach of Martinez-Boti, so the best comparison is with that study.

Yes, therefore it is even more surprising that we find higher values for ESS, when considering nearby time intervals as Royer (2016). To clarify, we have added (Line 275), “even though we use a similar approach (long-time average) as ref.⁵⁶.”

Lines 257-259: It's important to say here, in the main text, that your ECS is not the same as the ECS's in the IPCC; most importantly, you don't consider the potential for changes in other GHGs. You also don't consider changes in paleogeography or solar luminosity (short-term wiggles or long-term secular increase).

We agree. We have added Line 292-297, “It should be noted that our ECS is not the same as the ECS used by the IPCC, given that it represents specific climate sensitivity $S_{[CO_2,LI]}$ (i.e., ESS corrected for potential slow land ice feedback) and does not consider changes in other greenhouse gases (e.g., methane), paleogeography, nor solar luminosity; we are currently unable to conduct these additional considerations⁶⁶. The impact of additional methane and water would bring down ECS, which likely explains why paleo ECS is generally higher than modern models.”

I appreciate the efforts of the authors to now include the Rae et al. 2021 CO₂ compilation, but even if this manuscript had been submitted to another journal beforehand, when the manuscript was submitted to Nature Communications the Rae et al. 2021 paper was published, well known and should have been included in the original submission.

We agree with the reviewer. As explained previously, our manuscript came back from a very long review and unfortunately was directly declined. At that time, as the corresponding author was moving from her PhD position in the Netherlands to a postdoc position in the UK, it was decided to only make some minor modifications and to resubmit quickly, in view of the timeliness of the manuscript. However, the reviewer is right that at that time the conclusions of the Rae et al. compilation should have been taken into account in submitting the work to a new journal and we apologize for that omission.

In the new introduction, lines 22-39, they include Rae et al. but still start from an argument of the Foster et al. compilation and the ‘problem’ of low middle Miocene estimated CO₂ concentrations. In my view they still need to accept that the Foster compilation is superseded and that there’s no longer a significant middle Miocene problem. If they did this, I think they could make a better argument for the novelty of their approach and its importance – I think it’s more significant as a new method that can extend CO₂ records further back in time than the alkenone CO₂ proxy (and a study that tests the robustness of this method). I appreciate the work that has gone into the modern calibration and ground-truthing of this new CO₂ proxy approach, and also the detailed organic geochemistry to carefully extract and analyse these GPBs from DPDP 467 sediments – including the desulfurization steps – that are a significant benchmark and reference for such analyses. With more of a refocus on the importance (a revised introduction – see comments above) and details of the new methods (see comments below), I think this could be considered for publication.

We have removed the Foster et al. compilation from the manuscript. We have also made a note of the even more recent pCO₂ compilation in Science (Line 37-39), “the pCO₂ compilation by Rae et al.⁷ which provides revised revisions to the boron-based and alkenone-based pCO₂ proxies, considered the most robust proxies among the Cenozoic pCO₂ compilation⁸.”

To follow the reviewer’s suggestion regarding the proxies, we have changed the title to “Continuous sterane and phytane δ¹³C record reveals a substantial pCO₂ decline since the mid-Miocene”. In the abstract, we added Line 9-10 “...here using the first general phytoplankton biomarkers (phytane and steranes) applied to the Miocene” and later Line 100-101 “Furthermore, GPBs have the potential to span the Phanerozoic, whereas alkenones are limited to the Cenozoic, which would extend E_p-based pCO₂ proxies by nearly ten-fold.” We have also added the raw data for the calculations into the main text and figures.

Given the novelty of these records, I would like to see more of the primary data in the main text. At present they go straight to a CO₂ record in their first figure – it looks good and makes sense considering the long-term Neogene climate evolution, but the fact that it arrives “fully formed” as the first figure in the paper prompts questions as to the underlying assumptions and data – is it really as good as it looks (and I have serious concerns about the SST data – outlined below)? I would like to see the primary d¹³C of the biomarkers, the long-term planktic foraminiferal d¹³C compilation they use, the sea surface temperature records and the calculated E_p all plotted in the main text. Some of these are plotted in the SI but not all. All these data combine to generate the final CO₂ record, and I would like to see the trends, magnitudes of change and variability in these records in order to understand the components that contribute to the final CO₂ record.

We have added a new figure in the main text (Fig. 1) to include all data used and refer in the main text how we arrive from these data record to the reconstructed pCO₂ presented in Fig. 2 (previously Fig. 1).

I would also like further justification of:

- The use of the global compilation of planktic foraminiferal d¹³C – how was the uncertainty on this propagated through into CO₂ estimates? How well is this expected to represent local conditions at the study site? How do California Margin seawater DIC d¹³C compare to the “global ocean” today?

Preferentially, $\delta^{13}\text{C}$ for DIC would be derived from planktic foraminifera from the same site or nearby site; however, this was not possible given the depositional conditions at this site (which were very favorable for organic matter but not for carbonate preservation). That said, modern values at the site closely reflect the global average (see map below).

The uncertainty in each individual parameter within all of the equations was considered in Monte Carlo simulations to propagate a distribution of uncertainty on each $p\text{CO}_2$ estimate. The details of this methodology (along with the Python code) are available in Witkowski et al. (2018). The uncertainty of 0.4 per mil for the $\delta^{13}\text{C}$ of planktic foraminifera was based on analytical instrumental error of 0.2 per mil which was doubled for the consideration of the location-based uncertainty; this 0.4 per mil was then propagated throughout the Monte Carlo run equations.

- SST records - these should be plotted in the main text. How sensitive is the final CO_2 estimate to the choice of GDGT SST proxy? E.g. does the use of BAYSPAR substantially alter CO_2 records? I would like to see a sensitivity test with CO_2 curves plotted for at least the BAYSPAR SST estimations as well as the Kim et al. (2010) calibration of TEX₈₆.

The SST records are now plotted in the main text.

We include a $\pm 4^\circ\text{C}$ uncertainty on SST throughout the equations (see previous comment on $\delta^{13}\text{C}$ of DIC). $p\text{CO}_2$ sensitivity tests to temperature are described in detail in the Supplement of Witkowski et al. (2018). This temperature component has been a fundamental part of the fractionation-based $p\text{CO}_2$ proxy since its inception in the late 1980s, most importantly in converting dissolved $\text{CO}_{2[\text{aq}]}$ into atmospheric $p\text{CO}_2$ via Henry's Law.

We have not included BAYSPAR calculations because the uncertainty difference between the two GDGT methodologies will fall well within this $\pm 4^\circ\text{C}$ uncertainty.

Checking the data tables I'm also not happy with two things:

1) The SST used for the CO_2 estimations (S1) say that they are Uk37 based temperatures not TEX₈₆ based temperatures. What is going on here? The calculations in the tables don't match the methodologies outlined in the text?

Thank you for pointing out this labelling mistake in the Supplemental Table descriptions. We did indeed use TEX₈₆ SSTs for reconstructing $p\text{CO}_2$, as described in the manuscript on Line 141-144: "Sea surface temperatures (SST) were calculated using the TEX₈₆ proxy based on the ratio of cyclopentane rings in glycerol dibiphytanyl glycerol tetraethers (GDGTs) in the same sediments as our GPBs (Fig. 1c; Table S8) and assigned an uncertainty of $\pm 4^\circ\text{C}$ SD caused by potential calibration errors."

We use U_K^{37} -based global SST compilation for climate sensitivity, Line 255-256: “ Δ SST is based on the U_K^{37} proxy for SST, which were compiled⁹ by latitude and hemisphere into 0.125 Ma bins⁹ (Fig. 3B) and linearly interpolated for the age of our sediments (Fig. 4).”

2) And further, the actual TEX₈₆ data is not available for 10 samples at the base of the record because GDGT abundances are too low. Where is this mentioned in the methods? And what did they do without TEX (although as noted in point 1 – it actually looks like it's Uk37 that is being used). If one of the global records for SST from Uk37 is being used (I can't see it being from the actual site – as there are no alkenones recorded in the older part of the section), then I am very concerned about circularity here – driving the CO₂ proxy with a global SST signal and then comparing it back against that same SST curve and finding a strong correlation.... This may not be what they are doing, but I'd like a very good explanation of what is happening here, as the methods in the text don't seem to match the data tables. As with the failure to present the Rae et al. 2021 compilation, the jump straight to a finished CO₂ curve in the main text with none of the underlying data clearly shown in the main text, and then when the working is checked, to find it doesn't match the stated methods in the paper, and further to discover substantial chunks of (GDGT SST) data missing, is very concerning.

We use site-specific TEX₈₆ for the pCO₂ reconstructions and published U_K^{37} records from a variety of globally disperse sites for understanding climate sensitivity, indeed to keep these two separate (see previous response).

We have clarified this TEX₈₆ issue on Line 799-805, “Because several minor isoGDGT were below the detection level in the deepest part of the studied section, it was not possible to obtain TEX₈₆ values. To accommodate for this, we compared the overall records from Site 467 to the TEX₈₆ values at Site 608 at the same latitude, as well as U_K^{37} values from the nearby Site 1010 (directly south of DSDP Site 467) and Site 1021 (directly north of DSDP 467); all four sites have near-identical SST values throughout the past 15 Myr, so we use these other sites to linearly extrapolate the several missing SSTs at Site 467.”

In the figure below, we show DSDP 467 proxy-derived temperatures in orange (U_K^{37} light orange, TEX₈₆ mid orange, TEX₈₆ extrapolated dark orange), alongside the same latitude TEX₈₆ Site 608 (purple) and for further comparison, the nearby Site 1010 (grey) and Site 1021 (blue).

- The 'b' term. Although this is based on modern calibration studies, which look to be convergent on a similar number, I'm surprised at the "sensitivity" test of a "1% variability" in b which gives only a "1% change in CO₂" (lines 152 -156). This 1% variability doesn't even span the difference between the two figures they quote for b in the text (168 and 170 per mil kg μ M-1). I think those estimating CO₂ from the alkenone-based proxies would be over-joyed to know the 'b' term for the coccolithophores to within 1%. In the case of the alkenones

it's clear that estimates of the 'b' term in the modern ocean do not translate into an accurate representation of the b term on even glacial-interglacial timescales (e.g. see discussion in Rae et al. 2021; Badger et al. 2020; Zhang et al 2019). In the case of the coccolithophore alkenone-producers, this variation in the b term is within relatively few species and just through the late Pleistocene. So, I am sceptical that a 1% variation in the b term is an appropriate sensitivity test for general (unconstrained) phytoplankton biomarkers on significantly longer (~15 Ma) timescales. Not only could there be significant evolutionary adaptation to long-term declining CO₂ concentrations (as seen in the coccolithophores – see Bolton, Stoll and many other papers), in the single site there are likely to be significant changes in dominant phytoplankton community, growth rates and size distributions. I do appreciate the efforts to test community change through the cross-comparison of the d13C data with biomarker indications for upwelling, but I don't agree with the assumption that the b term is well-constrained (<1% variation!) in this application or that it is stable on long-timescales.

We would like to stress that we do not use a range of 1% as our uncertainty for the 'b' term. Indeed, as stated by the reviewer, that would be far too small. We explicitly use the range 168 ± 43 SD ‰ kg μM^{-1} , which reflects a ~25% SD uncertainty. The lines "Sensitivity tests demonstrate that 1% change in b results in only a 1% change in pCO₂ estimation" means that if the average for "b" for these preserved algae was 1% different from the average 168 ‰ kg μM^{-1} , it would change the pCO₂ estimate by 1%; if it was 10% different, it would change the pCO₂ estimate by 10%; if it was different by 40% then it would change CO₂ by 40%; and so on.

We refer to the "b" factor as "one of the more-difficult-to-constrain parameters in our calculation" (Line 172) and would like to emphasize that we are one of the few research groups who actually include a "b" range in their uncertainty analyses, let alone include a "b" value uncertainty with such large ranges. Hence, we feel that we have addressed the uncertainty in the "b" factor as well as we can.

Paragraph from Line 191 onwards – sounds like confirmation bias. That you get the answer for CO₂ from a proxy that fits with what the models need to reach Miocene temperatures is not the way to reassure yourselves that you have a robust CO₂ proxy – or that the models are robust. Keeping clear blue water between these two is important to provide confidence that they are independent, and we are properly testing model behaviour with the proxy data. Getting the 'right answer' is no guarantee that one's methods are sound, or, in this case, that one has actually arrived at the correct answer.

We agree with the reviewer. To clarify this, we have added Line 201, "to put our results into context of the literature..." and Line 213, "...it is notable that our GPB-based pCO₂ estimates are consistent with the pCO₂ required by the majority of climate models..."

Line 213 – 215; why don't they do the sensitivity test with at least two common approaches to GDGT-based SSTs? This would give at least some representation of what (minimum) SST uncertainty in their records might translate to in terms of CO₂ rather than relying on other studies. I also don't understand why they don't plot their own SST record from this site as well as the existing alkenone compilations? – the local SST records may not be the most helpful for the Fig 2 comparison of CO₂ against global SST records, but the fact that the data is not plotted anywhere is not helpful. See also the comments above about which SST they actually have from the site, and which data they use to calculate CO₂.

The SST data is plotted in the newly added Fig. 1. We are currently using SST uncertainty that is much larger than the uncertainty that would be derived from the suggested uncertainty test (see previous response). We do not use the U^K₃₇ records at this site in order to prevent circularity, given that we are also calculating global climate sensitivity in this manuscript (see previous response).

Finally, if I were Reviewer 1, I would not be satisfied that the revised manuscript has properly engaged with their comments and concerns on several fronts. One aspect that overlaps with my comments above is the discussion and presentation of the 'b' term. As reviewer 1 notes, we seem to be coming "full circle" and somehow finding that phytane CO₂ records are better constrained than the alkenone CO₂ records. I agree that they might be more widely applicable, but like Reviewer 1 I'm sceptical that d13C fractionation factors

from biomarkers that are more poorly constrained in terms of their biological source and growth environment can do better than ones that are well-constrained.

We thank the reviewer for this remark but Reviewer 1 states in their review report that “the authors have addressed many of my concerns” and this included the discussion and presentation of the ‘b’ term.

Reviewer #1 (Remarks to the Author):

The authors have addressed my concerns satisfactorily, thank you. Two small items:

Line 38: change to "considered the most robust proxies". It's not reasonable to judge what the two best proxies are---they all have their warts. If you read through the Cenozoic CO2PIP supplement, you wouldn't come away thinking that the alkenone proxy is the best (or second best) proxy.

ESS paragraph at the bottom of p. 10: Your average ESS of 13.9 oC is not too different from the Cenozoic CO2PIP estimate---if you look at their Figure 3, the average ESS from 20 Ma to present definitely exceeds 8 oC. I think this is worth saying. In contrast, the Wong et al. estimate (your ref. 57) is not the best comparison to your study because they made a single estimate for the entirety of the last 400+ Myrs; so I suggest cutting (and, for what it's worth, during times with ice sheets Wong assumes a 2X ESS, so for the Miocene you should double the 3.4 oC estimate that you see in their abstract).

Reviewer #2 (Remarks to the Author):

I thank the authors for their continued engagement with reviewers' comments and their extensive work on this manuscript. With the additional clarifications, details and revisions provided, I think it's in a state for the wider community to take a view and recommend publication. The use of the sterane and phytane approaches should be picked up by other groups, further tested in the modern, as well as in paleorecords from a range of sites and across timescales. Hopefully it will be found to be robust and provide a valuable addition to the critical efforts to understand past atmospheric CO2 variations.

Some minor points:

Lines 68 – 71: I don't understand the use of the word "momentous" in this context. Adding an irradiance control on Ep could be viewed as making a complicated and difficult-to-constrain b term, even more difficult. I don't think I'd greet this as momentous.

Also, the second part of this passage is troubling – it appears to be arguing that if we get the answer we're looking for all is fine, if not we can weed out the problems. I don't think this is a good approach to such an important question as pCO2 estimation – and I think sadly leads to a) confirmation bias of studies only being published that meet the "expectation" (a significant problem); and b) is not a good approach when parameters within the proxy system are poorly constrained – the risk is that these are chosen in different cases and different studies to achieve a 'good fit' to expectation, and so hinders a proper robust use and evaluation of a proxy under the same assumptions between studies and through time – otherwise they really are not the same "proxy". Properly, the uncertainty on poorly constrained parameters should be modelled in the deployment of the proxy, as is done in this study.

I think I see what they are trying to say - we need a diversity of independent proxy methodologies, ideally with consistent deployment of methods between groups, with honest and robust modelling of uncertainty within each system, to come to any confidence in the actual trend in atmospheric CO2 through time. In this context, we can then challenge and scrutinize persistent proxy outliers that

range outside the uncertainty bands of multiple other proxies.

Line 73 – “few Haptophyte species” – maybe a few in the modern, but across the Cenozoic, there are many tens of named Noelaerhabdaceae species (even back to the mid-Miocene). Given the occurrence of modern-type alkenones back to the Eocene, the conservative assumption would be that alkenone-production is a common feature of most of the family. And, given (pseudo-)cryptic diversity in modern species, the number of named fossil morpho-species is likely a very large underestimate of 'actual' species richness. Alkenone production is thus a feature of very many species (within a well-defined clade) in the Cenozoic, not a few.

Line 175 - "Sensitivity tests demonstrate that a 1% change in b results in only a 1% change in pCO₂ estimation³⁶³⁶, which is too small to account for the consistent decline over the studied time interval."

I'm going back to this as I still don't like the formulation. I see the authors response to this - and I get what they say. But in their calculations they allow for 25% variation in b, which would be associated with 25% uncertainty in CO₂, and as shown on their plots. 25% is the uncertainty in b in the modern, so I really don't understand why they use an example of a "1% change in b". OK, so a 1% change in b is associated with a 1% change in CO₂, but then see how that statement stands up with, "our uncertainty on b is +-25%". And that's in the modern system, in relatively few studies. I'd be more cautious of how, when and where b might vary significantly, and I don't think talking about a 1% variation in b is an appropriate estimate of the potential changes that there might be in this parameter through their record.

Reviewer #1 (Remarks to the Author):

The authors have addressed my concerns satisfactorily, thank you. Two small items:

Line 38: change to “considered the most robust proxies”. It’s not reasonable to judge what the two best proxies are---they all have their warts. If you read through the Cenozoic CO2PIP supplement, you wouldn’t come away thinking that the alkenone proxy is the best (or second best) proxy.

We have removed “robust”.

ESS paragraph at the bottom of p. 10: Your average ESS of 13.9 oC is not too different from the Cenozoic CO2PIP estimate---if you look at their Figure 3, the average ESS from 20 Ma to present definitely exceeds 8 oC. I think this is worth saying. In contrast, the Wong et al. estimate (your ref. 57) is not the best comparison to your study because they made a single estimate for the entirety of the last 400+ Myrs; so I suggest cutting (and, for what it’s worth, during times with ice sheets Wong assumes a 2X ESS, so for the Miocene you should double the 3.4 oC estimate that you see in their abstract).

We have added, “The Cenozoic CO2PIP⁷ estimations for ESS exceed ca. 8°C per CO2-doubling for the past 20 Myr, reaching ca. 13°C per CO2-doubling in the early Cenozoic.”

Reviewer #2 (Remarks to the Author):

I thank the authors for their continued engagement with reviewers' comments and their extensive work on this manuscript. With the additional clarifications, details and revisions provided, I think it's in a state for the wider community to take a view and recommend publication. The use of the sterane and phytane approaches should be picked up by other groups, further tested in the modern, as well as in paleorecords from a range of sites and across timescales. Hopefully it will be found to be robust and provide a valuable addition to the critical efforts to understand past atmospheric CO2 variations.

Some minor points:

Lines 68 – 71: I don’t understand the use of the word “momentous” in this context. Adding an irradiance control on Ep could be viewed as making a complicated and difficult-to-constrain b term, even more difficult. I don’t think I’d greet this as momentous.

We have changed “momentous” to “timely” to provide a more neutral term.

Also, the second part of this passage is troubling – it appears to be arguing that if we get the answer we’re looking for all is fine, if not we can weed out the problems. I don’t think this is a good approach to such an important question as pCO2 estimation – and I think sadly leads to a) confirmation bias of studies only being published that meet the “expectation” (a significant problem); and b) is not a good approach when parameters within the proxy system are poorly constrained – the risk is that these are chosen in different cases and different studies to achieve a ‘good fit’ to expectation, and so hinders a proper robust use and evaluation of a proxy under the same assumptions between studies and through time – otherwise they really are not the same “proxy”. Properly, the uncertainty on poorly constrained parameters should be modelled in the deployment of the proxy, as is done in this study.

I think I see what they are trying to say - we need a diversity of independent proxy methodologies, ideally with consistent deployment of methods between groups, with honest and robust modelling of uncertainty within each system, to come to any confidence in the actual trend in atmospheric CO2 through time. In this context, we can then challenge and scrutinize persistent proxy outliers that range outside the uncertainty bands of multiple other proxies.

We agree with the reviewer. We have changed the second part of this passage to, Line 53-56, “By using a diversity of independent proxy methodologies, ideally with consistent deployment of methods between groups, with honest and robust modelling of uncertainty within each system, we can then challenge and scrutinize persistent proxy outliers that range outside the uncertainty bands of multiple other proxies.” which more accurately reflects what we were trying to express in the text.

Line 73 – “few Haptophyte species” – maybe a few in the modern, but across the Cenozoic, there are many tens of named Noelaerhabdaceae species (even back to the mid-Miocene). Given the occurrence of modern-type alkenones back to the Eocene, the conservative assumption would be that alkenone-production is a common feature of most of the family. And, given (pseudo-)cryptic diversity in modern species, the number of named fossil morpho-species is likely a very large underestimate of 'actual' species richness. Alkenone production is thus a feature of very many species (within a well-defined clade) in the Cenozoic, not a few.

We have changed this to “compounds produced by species within the Haptophyte clade”. Given there is no definitive amount, we agree it is prudent to remove “few” (as we had originally written) and also prudent to exclude “many” (as the reviewer suggests).

Line 175 - "Sensitivity tests demonstrate that a 1% change in b results in only a 1% change in pCO₂ estimation³⁶³⁶, which is too small to account for the consistent decline over the studied time interval." I'm going back to this as I still don't like the formulation. I see the authors response to this - and I get what they say. But in their calculations they allow for 25% variation in b, which would be associated with 25% uncertainty in CO₂, and as shown on their plots. 25% is the uncertainty in b in the modern, so I really don't understand why they use an example of a "1% change in b". OK, so a 1% change in b is associated with a 1% change in CO₂, but then see how that statement stands up with, "our uncertainty on b is +-25%". And that's in the modern system, in relatively few studies. I'd be more cautious of how, when and where b might vary significantly, and I don't think talking about a 1% variation in b is an appropriate estimate of the potential changes that there might be in this parameter through their record.

We have changed this to “Sensitivity tests demonstrate that the uncertainty within the b value could lead up to a maximum of 25% change in pCO₂ estimation.”